# Unbridled Integrons: A Matter of Host Factors

**DOI:** 10.3390/cells11060925

**Published:** 2022-03-08

**Authors:** Egill Richard, Baptiste Darracq, Céline Loot, Didier Mazel

**Affiliations:** 1Institut Pasteur, Unité Plasticité du Génome Bactérien, UMR3525 CNRS, 75015 Paris, France; egill.richard@pasteur.fr (E.R.); baptiste.darracq@pasteur.fr (B.D.); 2Collège Doctoral, Sorbonne Université, 75005 Paris, France

**Keywords:** integron, bacterial genetics, mobile elements, site specific recombination, host factors, bacterial evolution, antibiotics resistances, horizontal gene transfer

## Abstract

Integrons are powerful recombination systems found in bacteria, which act as platforms capable of capturing, stockpiling, excising and reordering mobile elements called cassettes. These dynamic genetic machineries confer a very high potential of adaptation to their host and have quickly found themselves at the forefront of antibiotic resistance, allowing for the quick emergence of multi-resistant phenotypes in a wide range of bacterial species. Part of the success of the integron is explained by its ability to integrate various environmental and biological signals in order to allow the host to respond to these optimally. In this review, we highlight the substantial interconnectivity that exists between integrons and their hosts and its importance to face changing environments. We list the factors influencing the expression of the cassettes, the expression of the integrase, and the various recombination reactions catalyzed by the integrase. The combination of all these host factors allows for a very tight regulation of the system at the cost of a limited ability to spread by horizontal gene transfer and function in remotely related hosts. Hence, we underline the important consequences these factors have on the evolution of integrons. Indeed, we propose that sedentary chromosomal integrons that were less connected or connected via more universal factors are those that have been more successful upon mobilization in mobile genetic structures, in contrast to those that were connected to species-specific host factors. Thus, the level of specificity of the involved host factors network may have been decisive for the transition from chromosomal integrons to the mobile integrons, which are now widespread. As such, integrons represent a perfect example of the conflicting relationship between the ability to control a biological system and its potential for transferability.

## 1. Introduction

Mobile genetic elements (MGE) were originally considered as selfish and parasite DNA sequences. However, they are now viewed as genuine drivers of the evolution of their hosts. Over the course of their coevolution, in many examples, the function of MGE has become increasingly integrated into the physiology of the host, thereby both diminishing their potential deleterious effect and coupling their function to specific life stages. This integration may be achieved through the involvement of host factors (HF), which become key molecular players at the interface between the host and MGE functioning. Indeed, differential regulation of host factors, or of their activity, in response to changing environmental and physiological conditions will cause substantial alterations in MGE activity. These HF give to the host the power to regulate, to variable extents, the activity of the MGE. Moreover, in bacteria, the degree of host factor specificity may directly influence the potential of MGE dissemination by horizontal gene transfer (HGT), and their success among new host organisms. However, the strong selective advantage for a partnership between MGE and their hosts, has a downside by leading to a constrained interdependence. In this review, we will focus on a particular MGE, the integron system, and its interconnections with its bacterial hosts. Indeed, after more than 30 years of functional study of these very successful bacterial MGE, it is now established that they have explored a number of possible physiological controls, which have led to the selection of two types of integrons, “mobile integrons” (MI) and “sedentary chromosomal integrons” (SCI), which have distinct HF requirements and thus play different roles for their bacterial hosts.

## 2. The Unique Integron Genetic System

Integrons are bacterial recombination systems acting as platforms capable of capturing, stockpiling, reordering, and expressing DNA sequences embedded in promoterless cassettes. The type of integrons that were the first discovered are the MI [1]. These integrons are “mobile” since they are always associated with transposons and carried on conjugative plasmids. They contain mostly antibiotic resistance cassettes and remain the best studied integrons [2]. This is justified when considering the ever-increasing antibiotics pressure around the world [3] and the over-representation of integron-containing bacteria in the clinical context, where they cause 22% to 59% of infections [4]. Multi-resistance to antibiotics in bacterial pathogens is already one of the biggest threats to public health, and the integron is one leading cause of such phenotypes [5]. The reason for this is that integrons allow cells to concentrate resistance genes to many different antibiotics while limiting the cost associated with the presence of a resistance gene in the absence of the ad hoc antibiotic [6]. Apart from antibiotic resistance genes, integron cassettes can encode many other functions, especially within the other type of integrons carried by chromosomes and so-called sedentary chromosomal integrons. SCI are found in approximately 10% of all currently sequenced genomes [7], particularly in almost all *Vibrio* species [8]. The functions found in SCI probably allow the bacterium to respond to stresses that are relevant to its lifestyle playing a role in bacterial evolution [9].

Both MI and SCI share the same general organization: a stable platform and a variable array of cassettes (Figure 1). The stable platform of the integron contains (i) the integrase gene (*intI*) under the control of its promoter P_int_, (ii) the *attI* integration site, and (iii) the cassette P_C_ promoter driving the expression of the genes encoded in the cassette array located downstream. The variable part consists of an array of cassettes, each of which are generally composed of a promoterless gene associated with a recombination site called *attC*. Only the first few cassettes (those closest to the P_C_ promoter) can be expressed, while the rest represent a low-cost memory of valuable functions for the cell [10]. Upon expression of the integrase, the cassettes can be excised (*attC* × *attC* recombination events, Figure 1) and then re-integrated at the *attI* integration site (*attC* × *attI* recombination events), thus becoming expressed. The combination of excision and integration of cassettes, i.e., their shuffling, allow bacteria to screen for the set of functions that might optimize their survival in a given environment.

The integron integrase has a very singular place among the broad family of the tyrosine recombinases. Indeed, while it recognizes the *attI* site under its double-stranded (ds) form through its primary sequence (Figure 2A), this is not the case for the *attC* sites (Figure 2B). *attC* sites show a very limited conservation in sequence but all share an imperfect palindromic organization that allows for the formation of a single-stranded (ss) secondary structure that is recognized and recombined by IntI (Figure 2B). Although both the bottom and the top strand of an *attC* site can form a secondary structure, the recombination of the bottom strand (bs) is about 10^3^ more efficient, and the structural features of the site play a central role in this strand selectivity [11,12,13]. The imperfections in the palindromic organization of the *attC* sites drive specific structural features upon folding (Unpaired Central Spacer (UCS), the Extra Helical Bases (EHB), and the Variable Terminal Structure (VTS)) which direct the strand specificity toward bs binding and recombination [12]. This selectivity for the bottom strand of the structured *attC* site is essential for the correct orientation of the cassette upon integration at the *attI* site, allowing for its expression by the P_C_ promoter [14].

In this review, we will focus on the extensive and exquisite interconnectivity of the integron system. We will particularly focus on the host factors that bridge their functioning to the biology of the host. We will first examine how the expression of the cassettes is connected to the host cell physiology. We will then detail how the observed integrase expression regulation is relevant in both natural and clinical contexts. We will also discuss the mechanistic aspects of their recombination reactions, considering the host biological processes and molecular actors involved to achieve those recombination reactions. Finally, we will examine how any interconnection between the integron and the host comes at the cost of constraints in terms of dissemination. This apparently led to the emergence of the MI quasi species, which probably owe their high success to their limited need of specific HF, but which, however, rely on the much more controlled SCI cassette reservoir to provide them with an unlimited source of cassettes to be able to spread in an unbridled fashion.

## 3. Host Factors Influencing the Expression of the Cassettes

The vast majority of the integron cassettes do not carry any promoter, so their expression depends on the P_C_ promoter that is upstream of the cassettes. This promoter drives the expression of the cassettes according to a gradient so that expression levels are maximal for the first cassettes in the array, and they gradually decrease for those following, while the distal ones are not expressed [10,15,16,17]. This feature is essential to maintain large arrays of cassettes at the lowest possible cost. This is particularly true for SCI, in which the array of cassettes can be gigantic. The SCI paradigm is that of *Vibrio cholerae*, termed Superintegron (SI), which contains about 180 cassettes. The expression of cassettes has been thoroughly studied in this species, and the corresponding P_C_ promoter is well characterized [18]. In particular, a CRP-cAMP binding motif [19] can be found in vicinity of the -35 box of the SI P_C_ promoter (Figure 3). This allows one to connect the expression of the cassettes to the host metabolism through the catabolic repression. It was found that in the SI, the cassette expression is not constitutive and that the activity of the P_C_ promoter is rather modulated by several factors, including the carbon source (low expression in presence of glucose and high expression in presence of succinate or arabinose) and the growth phase (low expression in exponential phase and high expression in stationary phase [20]). Interestingly, this means that CRP can integrate many metabolic signals, through its binding to cAMP and subsequent activation of P_C_ activity, to influence the range of cassettes that can be expressed in the SI. In conditions of metabolic stress that increase the cAMP levels in the cells, the gradient of expression of the SI cassettes will be extended, possibly favoring the expression of the appropriate combination of cassettes that will allow the cell to cope with the initial stress. Other factors such as temperature and salinity were also found to modulate cassette expression, but this is out of the scope of this review since we cannot assign such general stresses to a particular host factor, especially considering the fact that the general stress response regulator gene *rpoS* [21] was shown not to be involved in this process.

P_C_ promoters of 3 of the 5 MI classes have been also characterized. The class 2 and 3 P_C_ promoters have a limited characterization [22,23], but the class 1 P_C_ has been more extensively studied. Class 1 integrons are the main resistance cassette vectors in Gram negative bacteria, and they were found to carry many different variants of P_C_ [24], which have various strengths and confer different recombination properties to the IntI. Indeed, as the Pc promoter is embedded in the *intI1* gene, the Pc variants lead to differential amino acid substitutions within the *intI1* gene. This promoter has been found to be regulated by one nucleoid-associated protein (H-NS [25]); no CRP box could be found in any of these MI P_C._ Thus, if there is an interconnection between the metabolic state of the cell and the expression of the cassettes in the MI, it is an indirect one, and the CRP control seems to be a particularity of the SCIs. This marks the first difference of connectivity between the host and SCI or MI. For more details concerning the P_C_ integron promoters, you can refer to the review from Fonseca and Vicente [26].

## 4. Host Factors Influencing the Expression of the Integrase

### 4.1. Integrase Expression and SOS Response

#### 4.1.1. The Integrase Gene Is Part of the SOS Regulon

The SOS response is an almost ubiquitous response of cells to genotoxic stresses, sensed as the accumulation of single-stranded DNA (ssDNA). The term “SOS response” was coined by Miro Radman in the 70’s and was originally associated with DNA damage [27] and later with HGT [28]. This pathway involves two main regulators: the LexA and RecA proteins. LexA is a specific transcriptional repressor [29]. The RecA protein acts as a detector of the “abnormal” presence of single-stranded ssDNA in the cell, which can come from double-strand break repair or arrested replication forks, as well as from the entry of exogenous DNA molecule by HGT [30]. RecA forms a nucleofilament with the ss DNA molecule and becomes activated to stimulate the LexA inactivation by autoproteolysis and subsequent degradation by ClpXP and Lon [31]. Inactivation of LexA leads to the derepression of all genes belonging to the SOS regulon. In *Escherichia coli*, this regulon includes 57 genes that code for proteins involved in DNA replication, repair, and recombination; cell division arrest; and translesional replication systems [32]. In *V. cholerae*, the SOS regulon overlaps the *E. coli* one but also includes several additional genes [33]. In absence of trigger, LexA forms a dimer and acts as a transcriptional repressor for genes belonging to the SOS regulon by binding to a specific operator sequence (the LexA-box) in their promoter region [34], which consists of a CTGT(N)_8_ACAG motif in gamma-proteobacteria [35]. The success of the integron system resides in the fact that the cassettes it contains can be shuffled until an optimal organization of the array is selected in a given period and environment context. If this is not the case, unneeded shuffling might do more harm than good to the cell. Indeed, as seen for many recombinases, the constitutive expression of the integrase over a long period of time can prove toxic for the cell [36]. This implies that the integrase expression has to be set “on” and “off”, depending on the need of the cell to get access to the diversity of functions encoded in the cassettes. In the case of integrons, like for other MGE [37], this is achieved through the integration of the integrase expression in the SOS response [38], thanks to the presence of a LexA box in the P_int_ promoters of many integrons integrases [39]. The expression of both integrases—the integrase of the SI of *V. cholerae* (IntIA, Figure 3) and of the class 1 MI (IntI1)—was analyzed. Induction of the SOS response increased the expression of a β-galactosidase reporter of integrase transcription 4.5-fold for IntI1 in *E. coli* and 37-fold for IntIA in *V. cholerae* [39].

Furthermore, there is a strong correlation between the LexA regulation and the functionality of the integrase: P_int_ lacking lexA box or with a poorly conserved LexA motif is often associated with a truncated, non-functional integrase [39]. In particular, in *Acinetobacter baylyi* (which lacks the *lexA* gene), the negative impact of the presence of an active IntI1 integrase is counterbalanced by the appearance of compensatory mutations that inactivate the protein during evolutionary experiments [40].

Moreover, it was shown that in class 2 MI, LexA is not able to bind to the P_int_ even if a potential LexA-binding box can be identified in this promoter [23]. Again, this lack of regulation of the expression of the integrase is associated the almost ubiquitous inactivation of IntI2 through non-sense mutations.

The regulation of the expression of the integrase by the SOS response is a highly conserved feature in integrons, and the few existing examples of SOS-independent expression of the integrase are associated with disruption of the functionality of the integron system. It is clear that the SOS-dependent expression of the integrase through the LexA repression is at the center of the integron functioning. This allows one to link the expression of the integrase with the presence of intracellular ssDNA, be it from genotoxic stress or from HGT.

#### 4.1.2. SOS Response and Antibiotic Induced Stresses

In most cases, ssDNA originates from double-strand breaks (DSB). Many factors can induce the formation of DSB in bacteria, whether due to external damaging agents or not. In absence of any damaging agent in the medium, it was shown that most of the DSBs stem from topological stress applied on the chromosome or by the formation of R-loops [41]. In the case of DSB due to such intrinsic factors, the SOS response is essential to efficiently repair the DNA damage and allow the cell to resume a normal lifestyle [27]. It was argued that, at lethal doses, the majority if not all antibiotics trigger the SOS response [42,43]. For the antibiotics directly targeting the synthesis of DNA (trimethoprim/sulfamethoxazole) and the maintenance of the genome (quinolones), this connection is straightforward. For instance, quinolones target topoisomerases of class II (DNA gyrases) and IV. Upon binding to their target, they will block their activity and therefore cause high topological stress that leads quickly to the accumulation of DSB, as mentioned previously. Therefore, quinolones such as ciprofloxacin are potent inducers of the SOS response [38,44]. Similarly, compounds that alter directly DNA, such as mitomycin that causes T-T cross-links, also efficiently trigger the SOS response [30,38]. For rifamycin, this blocks the activity of RNAP and therefore transcription; its side effect is to cause RNAP stalling and hence replication-transcription conflicts and formation of R-loops that both lead to high genomic instability [41]. The cascade of events linking the direct effect of the antibiotics and DNA damage can yet be more indirect. β-lactams, which inhibit the synthesis of the peptidoglycan and cause cell-wall damage, were shown to induce the two-component system DpiBA [45]. Induction of this signaling pathway ultimately leads to replication fork arrest and induction of the SOS response. Finally, many bacteria antibiotics targeting protein synthesis such as aminoglycosides are also able to trigger the SOS response [46,47]. Indeed, because of the coupling of translation and transcription in bacteria, the action of aminoglycosides on translation also alters the transcription rates and causes RNAP stalling, and amino acids miss-incorporation, leading to ROS production and to DNA damage [30]. The SOS regulon is an integrated pathway and will not be activated the same way, or with the same cues, in every species. For instance, as mentioned above, aminoglycosides generally trigger the SOS response in *V. cholerae* as well as in other species such as *Klebsiella pneumoniae* and *Pseudomonas luminescens* [48] but not in *E. coli*. Indeed, in *E. coli* a specific mechanism of control of the oxidative stress response prevents the SOS induction by nongenotoxic antibiotics [48].

Hence, the expression of the integrase upon induction of the SOS response is an ideal adaptive pathway for the cell in the context of antibiotics treatments. In their presence, if the integron contains the adequate resistance cassette but it is out of range of the P_C_ promoter, then the regulation of the integrase by the SOS response takes all its importance. The array will be rearranged at random and quickly generate important diversity in cassette order, on which selection can operate so that the best fitting configuration of array will become advantageous. Thus, resistant clones will emerge at a much higher rate than it would have while counting solely on the mutation rate. This is especially true in the case of combinatorial antibiotic treatments that completely forbid the appearance of resistant clones through mutations only. Integrons, because of their modular organization, allow the cell to adapt to multiple stresses, simultaneously providing adaptation “on demand” [36]. Sadly, this chain of events has been at the origin of a fatal treatment failure in a patient infected by *Pseudomonas aeruginosa*, where a silent resistance cassette was called in first position though integrase activity after treatment with another antibiotics, rendering the strain fully resistant to the last-resort antibiotic [49].

#### 4.1.3. SOS Response and Horizontal Gene Transfer

Another important source of ssDNA in bacteria is horizontally transferred DNA. HGT is a primordial aspect of bacterial evolution. HGT allows adaptation at a much higher pace through parallelized accumulation of adaptive traits [50]. For integrons, combination of metagenomics and comparative genomics approaches suggests that there is a very large pool of existing integron cassettes that can be found in a virtually infinite range of environments [51]. The only way those sets of cassettes coming from various backgrounds can be effectively compiled together within individual integrons is through HGT. Interestingly, in two of the most prevalent mechanisms of HGT in bacteria, conjugation, and transformation, the DNA is acquired in form of ssDNA [52].

During conjugation, a relaxase creates a nick in one of the strands of the plasmid DNA at the oriT. The nicked strand is then unwound from the unbroken strand and transferred to the recipient cell, which leads to the transient occurrence of ssDNA in the recipient cell [53]. This process has been visualized in real time and in single cells of *Escherichia coli* [54]. This triggers the SOS response and consequently allows the integrase to be expressed in recipient cells, as was shown in *E. coli* and *V. cholerae* [55]. One of the consequences is to induce cassette mobility within the host integron. In particular, in recipient *V. cholerae* cells, cassette shuffling within the SCI can be observed at quite high rates [56].

Whether in a clinical setting or in the environment, conjugation happens in high cell density conditions such as in biofilms or within the intestinal tract in the case of *V. cholerae*. In such contexts, conjugation can occur between very distantly related species [57]. As a consequence, a given MI on a conjugative plasmid can be quickly mobilized and encounter a very wide range of genetic backgrounds. If the species encountered by such an MI contain a SCI, it is easy to imagine that the sole conjugation-induced expression of the MI integrase could be sufficient to recruit useful functions from the SCI to the MI. It was shown, for instance, that IntI1 induction in a strain of *V. cholerae* containing the R388 conjugative plasmid could cause the recruitment of the *catB9* cassette from the SCI to the MI present in R388 [58]. Subsequent conjugation of that plasmid conferred resistance to chloramphenicol to the recipient *E. coli* strain. Hence, conjugation clearly is a highly relevant way to couple the mobility of integrons with their ability to confer extreme adaptability to their host.

Transformation is another important route for HGT, although it is restricted to the species that can develop competence in some specific conditions. Upon induction of competence, the mechanism of DNA uptake through transformation is relatively conserved in the different naturally competent species [59]. The exogenous dsDNA is the substrate for transformation: one strand is degraded, and the other is internalized through the ComEC transmembrane channel. Again, the result is that the newly acquired DNA is in its single-stranded form, inducing the SOS response in the recipient cell [20]. There are more than 80 species known to be naturally competent; *V. cholerae* is one of them, and likely most *Vibrio* species as their genomes are found to carry the competence genes [59]. It was found that *V. cholerae* acquires competence in presence of chitin, a component of the crustacean shells, which is coherent with the marine life-style of this species [60]. Similarly to what has been shown for conjugation, the chitin-induced uptake of exogenous DNA was sufficient to induce the expression of the integrase through the SOS [20]. In a recent report, for the first time, we provided evidences for cassette recruitment within the SCI of *V. cholerae* while relying solely on the SOS-dependent induction of IntIA [56]. Indeed, the efficiency of integration of a cassette provided as exogenous DNA in a chitin containing medium approached that of a cassette provided by conjugation without over-expression of IntIA (approximately 10^−6^). Deletion of the endogenous integrase did not allow for the observation of a single recombination event, proving that the induction of the endogenous integrase through natural transformation is enough to perform efficient recruitment in this natural setting. Interestingly, apart from the integration of the incoming cassette, the induction of IntIA through SOS response also caused cassette rearrangements within the SCI at a high frequency [56]. Natural competence has often been viewed as a stress response by itself, allowing the concerned organism to efficiently acquire exogeneous DNA as a way to adapt quickly to a new environment [61]. Hence, the interplay between the induction of competence and of the expression of the integrase in *V. cholerae* is highly illustrative of the adaptive potential of this species. In a context of stress, a cassette coming from the environment or from an integron-containing bacterium [62] can be efficiently be recruited within the SCI and possibly allow the host to escape the original stress. In line with this, at the multispecies population level of a given environment, Boucher and colleagues have found that cassette pools are rapidly transferred across species boundaries to generate an endemic population structure [63].

Finally, another major contributor of HGT is phage-mediated transduction. This pathway differs from conjugation and transformation because it can result from the injection of either dsDNA or ssDNA depending on the nature of the phage. However, the infection by a phage with a dsDNA genome can still lead to SOS induction through the disruption of the cell-cycle and the associated genotoxic stress.

There is only one example of formally characterized horizontal transfer of integron cassette by generalized transduction [64]. In *Salmonella enterica* serovar *typhimurium*, some strains are resistant to multiple antibiotics due to the presence of one or several integrons within their genome. Co-transduction of integron cassettes conferring simultaneous resistance to chloramphenicol, ampicillin, and tetracycline originating from the strain DT104 could effectively be observed in several other strains of the same species. However, it is very unlikely that these events correspond to cassette recruitment but rather to allelic exchange by homologous recombination with the transduced DNA.

### 4.2. Other Factors Influencing the Expression of the Integrase

#### 4.2.1. Catabolite Repression of the Integrase

As mentioned above, a CRP box is present upstream of the P_C_ promoter of the SCI in *V. cholerae* (Figure 3) [18]. This is also true in other species of *Vibrio*, so this might be characteristic of SCI in general [18]. Contrary to class 1 MI, in which P_C_ and P_int_ are facing each other, in SCI, they are back-to-back. Consequently, the CRP box that regulates the expression of SCI cassettes also regulates the expression of the integrase. Upon binding of the cAMP-CRP complex on its motif, the expression of the integrase is enhanced to reach a significant level and allow for cassette rearrangement. As mentioned above, the CRP regulon is activated when rapidly metabolizable carbon sources are scarce and can therefore be considered as a stress response just like the SOS response, except that the range of stresses that can induce it is much broader. In its natural aquatic niche, *V. cholerae* forms biofilms on the shells of crustaceans and mostly grows using chitin as a carbon source. In this adverse environment, where both the metabolic stress and the interspecies competitiveness are high, the induction of the CRP regulon and subsequent SCI rearrangements might provide a decisive adaptive advantage. Obviously, there is also a cross talk with the regulation of competence through the metabolization of chitin, especially since CRP induces the HapR quorum-sensing regulator that is another essential player in the induction of competence in *V. cholerae*. The synergy between the CRP regulation and HGT induced SOS response highlights the very high adaptive potential provided by SCI in environmental contexts. In contrast, CRP represses virulence factors of *V. cholerae* (the cholera toxin and the toxin-coregulated pilus), promoting its ability to colonize a medium rich in nutrient that is the human intestine. Hence, the catabolite repression of the integrase is illustrative of the bi-modal lifestyle of *V. cholerae* that alternates between a marine environment and a pathogenic context. The CRP regulation adds another layer of complexity that links plasticity of the SCI with multiple environmental cues. Importantly, this link does not exist in MI since, as said above, they are devoid of CRP boxes. We might argue that, by nature, MI are susceptible to be active in many different bacteria with many different lifestyles, while SCI are much more adapted to their host since they co-evolved for hundreds of millions of years [65]. The double regulation pathway of the SCI integrases represents the ability of the host to optimally exploit the “adaptive machines” that are SCI.

#### 4.2.2. Integrase Expression and Stringent Response

Finally, an important regulatory pathway of the integrase is the stringent response, which is another stress response in reaction to amino-acid starvation, heat-shock, and many other extreme conditions [66]. In particular, the stringent response is essential in the context of biofilms that are a prevalent lifestyle for most bacteria. In biofilms formed by *E. coli* carrying a class 1 MI, cassette excision events catalyzed by IntI1 were more frequent, and this was associated with the activation of the stringent response [67]. Indeed, in the context of biofilms, it was shown that the deletion of the *spoT*, *relA,* and *lon* genes decreases IntI1 expression. In their model, Strugeon and coll. propose that, within biofilms, the stringent response regulates integrase expression in two different ways. The first pathway depends on the interaction of the (p)ppGpp alarmone with RNAP, which can lead to transcription arrest and thus activate the SOS response. The second pathway involves the Lon-polyP complex, which is thought to act on an unknown regulator that interacts with the P_int_ promoter in the absence of LexA repressor binding to enhance *intI* expression. The fact that the integrase is induced in biofilms through stringent response is an important feature of integrons. It synergizes with both the SOS and the CRP regulations, such that the integrase expression might peak in this condition where HGT is maximized. Considering the prevalence of this lifestyle in the environment and the fact that class 1 MI are regularly found within bacterial biofilms [68], it is possible that they shelter a major route for the dissemination of antibiotic resistance [69].

## 5. Host Factors Influencing the Recombination of the Single-Stranded *attC* Substrates

As mentioned above, integron integrases recombine the *attC* sites under a structured ssDNA form. This offers another level of regulation and involves the possible involvement of host factors in the *attC_bs_* structuration. In addition, during integron cassette insertion, also due to the single-stranded nature of the *attC* sites, the generated Holliday junction is resolved by replication in a host-dependent manner.

### 5.1. attC Site Folding Regulation

DNA can take on a variety of secondary structures formed from more or less degenerate palindromic sequences. Secondary structures have been identified in the genomic DNA of archaea, prokaryotic, and eukaryotic species, including humans, and play essential roles in various DNA transactions [70]. Their structuration can occur by two main mechanisms, i.e., (1) the single-stranded pathway, which generates a hairpin structure; and (2) the double-stranded (ds) pathway, which consists in the extrusion of a cruciform structure from ds DNA (Figure 4). We have shown that, depending on the nature of the *attC* sites, both ways can be used to produce recombinogenic substrate. However, each of these two pathways involves the intervention of several host factors that may differ.

#### 5.1.1. Host Factors Involved in the *attC* Site Folding during the Single-Stranded Pathway

In several bacterial cellular processes, DNA can be single-stranded, for instance, during replication; during DNA repair; or, more importantly, during rolling-circle replication, bacterial conjugation, natural transformation, and infection by some viruses. We previously investigated some of them, such as conjugation, natural transformation, and replication, and demonstrated that they favor the proper folding of *attC* sites. During conjugation, ssDNA is unwound from the duplex plasmid, actively transferred into a recipient neighboring bacterium, and recircularized (for review, see [71]). We demonstrated that, by this way, conjugation ensures the folding of *attC* sites and that most of the *attC* sites displayed similar recombination frequencies when delivered by conjugation, regardless of their size or VTS length. Indeed, VCR sites with large VTS and *attC* sites with shorter VTS (e.g., *attC_aadA7_*) are recombined with similar efficiency (Around 10^−3^, [72]). On the contrary, in replicative recombination conditions, a significant correlation was observed between the recombination frequencies and the VTS length of *attC* sites (i.e., up to 4-log of decrease using *attC* sites with long VTS, [72]). The cassette recruitment is also facilitated by the ssDNA delivery during the transformation process [56]. Up to now, it was demonstrated that natural transformation mediates uptake and exchange of free ssDNA when sufficient homology is present between the incoming DNA and the bacterial genome [73]. We demonstrated that integration of genetic material during natural transformation can be also mediated by homology independent DNA mechanisms, notably by recombining gene cassettes inside integrons in an integrase dependent way. Hence, HGT not only contributes to the transfer of cassettes, it is also a perfect route for recruitment by integrons simply because it favors the folding of recombinogenic *attC* sites. This allows one to couple cassette recombination in bacteria with the moment of their entry under a single-stranded DNA form.

Another source of single-stranded DNA in the cell is the replication process. We also demonstrated that replication favors *attC* site folding [72]. During replication, one of the two new strands, the lagging strand, is made as discontinuous DNA pieces, and the corresponding template strand contains large single-stranded regions (i.e., between Okazaki fragments). We demonstrated that when the bottom recombinogenic strand of the *attC* site (*attC_bs_*) is located on the lagging strand template, its folding is favored. This has been proven by cloning an *attC* site in both orientations toward replication in a unidirectionally replicated plasmid and comparing their respective integration rate. The results showed that higher availability of ssDNA in the lagging strand template impacts the frequency of *attC* × *attI* recombination [72]. We also showed that the orientation of the cassette array relative to replication regulates the cassette excision rate (i.e., *attC* × *attC* recombination), by influencing the *attC* site folding [74]. Interestingly, in silico analyses reveal that *attC_bs_* in SCI are predominantly carried on the leading strand template and that there is a correlation between the cassette array orientation and its respective size [74]. Indeed, the few SCI identified in inverse orientation, found exclusively in *Xanthomonas* species, present a number of cassettes that do not exceed 22 cassettes, while those found in *Vibrio* in the opposite orientation can reach up to 200 cassettes (217 in *Vibrio vulnificus*). This bias of orientation in SCI, by limiting the cassette excision rate, could avoid cassette loss and ensure a large reservoir of genetic functions that will stay accessible for their MI counterparts.

On the other hand, the *attC* sites could be a target for several host factors and more precisely for the proteins able to bind single-stranded DNA such as the *E. coli* Single-Stranded DNA-Binding protein (SSB). SSB regulates many DNA processes such as replication, reparation, and homologous recombination [75]. SSB can spontaneously migrate along ssDNA. This diffusion helps in the local displacement of SSB by an elongating RecA filament. SSB diffusion also melts short DNA hairpins transiently and stimulates RecA filament elongation on DNA with secondary structures [76]. We precisely demonstrated that SSB is able to strongly bind folded *attC* sites and to destabilize them. The SSB effect is observed only in absence of the integrase. Indeed, when the integrase is expressed, the SSB overexpression or inactivation do not impact the recombination reaction mediated by the integrase [77]. Thus, the integrase is able to counterbalance the observed effect of SSB on *attC* site folding. The integrase possesses an intrinsic property to capture *attC* sites at the moment of their extrusion, stabilizing them and recombining them efficiently (Figure 5). The precise mechanism of action of SSB was elucidated using an in vitro approach based on single-molecule Förster Resonance Energy Transfer. The results showed that *attC* hairpins have a conserved high GC-content near their apical loop, enabling SSB to open the hairpin and create a dynamic equilibrium between *attC* fully opened by SSB and a partially structured *attC*-6–SSB complex. This latter complex is recognized by the integrase IntI, which extrudes the full hairpin upon binding while displacing SSB [78]. During this process, the SSB-flattened *attC* sites are converted in IntI-bound recombinogenic forms. In conclusion, we demonstrated that SSB, which hampers *attC* site folding in the absence of the integrase, likely plays an important role in maintaining the integrity of the *attC* sites and thus the recombinogenic functionality of the integron *attC* sites (Figure 5). Indeed, the stability of DNA secondary structures in the chromosome must be controlled and restrained. Too stable secondary structures could be deleted by slippage or by action of the SbcCD complex, which cleaves DNA hairpins [70]. Moreover, such structures could be toxic for the cell by blocking the replication progression [79].

#### 5.1.2. Host Factors Involved in *attC* Site Folding during the Double-Stranded Pathway

*attC* sites can be folded from dsDNA. Indeed, inverted repeats (perfect or imperfect) have the potential to form branched structures called cruciforms, in which inter-strand base pairing within the symmetric region is replaced by intra-strand base pairing [80]. The formation of a cruciform involves a great deal of structural disruption as it requires a complete reorganization in base pairing. We demonstrated that ability of *attC* sites to be extruded depends on two parameters: first, the length of their VTS since to go from a ds state to a cruciform, an *attC* site needs to melt at least the length of its VTS; and second, their propensity to form non-recombinogenic structures [72]. In summary, only *attC* sites with small VTS (e.g., *attC_aadA7_*, *attC_ereA2_*…) and low propensity to form non-recombinogenic structure could be efficiently extruded. In addition to these two parameters, we demonstrated that DNA superhelicity influences the *attC* site extrusion in vitro and in vivo [72]. Indeed, cruciforms have lower thermodynamic stability than the “classic” double-stranded DNA. They have been observed only in negatively supercoiled molecules, in which the unfavorable free energy of formation is offset by the topology of the torsionally stressed molecule. In agreement with these observations, we demonstrated that proteins increasing negative superhelicity such as gyrase and topoisomerase directly favor cruciform extrusion of *attC* sites and their recombination [72]. Precisely, we demonstrated that in a *topA10 gyrB266* strain, *attC* sites recombine about 10 times less efficiently than in a WT strain. By extension, these results mean that factors expected to influence the superhelicity in bacteria could modify the rate of cassette recombination. Replication, transcription, mutations in certain genes (such as those encoding RNA polymerase), HU, HNS, salt shock, high osmolarity, nutrient downshift, infections by phages and late phase of growth: all these factors influence the local superhelicity [81,82,83,84]. The level of superhelicity can also vary between bacterial strains. Indeed, the average supercoil density of a pBR reporter plasmid extracted from mid-log cultures of *Salmonella* is 13% lower than that from *E. coli* [85]. Interestingly, topology analysis of reporter plasmids also revealed higher levels of negative supercoiling in strains with the constitutively expressed SOS network, suggesting a link between the induction of SOS response and changes in DNA topology [86]. The network of *attC* site regulation linked to the superhelicity is therefore considerable and has not yet been completely explored.

### 5.2. Resolution of the Atypical Holliday Junction

#### 5.2.1. The Replication Process Is Involved in the aHJ Resolution

Site-specific recombination catalyzed by tyrosine recombinases follows an archetypical pathway consisting of two consecutive strand exchanges. The first strand exchange generates a Holliday junction, which is resolved by a second strand exchange. As said above, integron integrases are atypical Y-recombinases since they recognize and recombine double-stranded *attI* site with a folded single-stranded *attC* site, the bottom one. A consequence of this *ss attC* specificity is that the *attI* × *attC* reaction, which leads to cassette insertion, involves both dsDNA (*attI*) and ssDNA (*attC*), thus resulting, after the first strand exchange, in the formation of an atypical Holliday junction (aHJ) (Figure 6). Due to the asymmetry of this complex, a second strand exchange on the *attC* bottom strand would form linearized abortive recombination products. We demonstrated that aHJ resolution would rely on a mechanism involving the host machinery of replication [87]. Using an *attC* site carried on a plasmid with each strand specifically tagged, we followed the destiny of each strand after recombination. We demonstrated that only one strand, the one carrying the *attC* recombinogenic bottom strand, is exchanged. Indeed, the recombination products contain the *attC* site bottom strand and its entire de novo synthesized complementary strand. Therefore, we demonstrated the replicative resolution of single strand recombination in integrons and ruled out the involvement of a second strand exchange of any kind in the *attC* × *attI* reaction.

Note that the nature of host factors is not yet completely known because we do not know if the resolution occurs by the passage of the replication fork and/or by the intervention of host factors restarting stalled replication forks such as the PriA helicase [88]. We also studied the recombination between two *attI* sites, a reaction involving exclusively double-stranded substrates, for which the second exchange of strands is not abortive. Using genetic approaches and high-throughput sequencing, we demonstrated in vivo that both strands of the *attI* site are reactive and prove that the resolution of this HJ can follow either the replicative pathway distinctive of the integron integrase (see above) or the second strand exchange characteristic of Y-recombinases [89].

Altogether, these results demonstrated that integrases have evolved towards new recombination substrates (i.e., single-stranded) and resolution pathways (i.e., replicative way) connecting the integron with the cell physiology.

#### 5.2.2. The RecA Protein Would Be Involved in aHJ Resolution

We observed a role of RecA in the cassette recruitment in the SCI of *V. cholerae*. Indeed, we showed that cassette recombination frequency decreases by about 2 orders of magnitude in the *recA* deleted strain, suggesting an activator role of RecA in *attC**_aadA7_* × *attIA* recombination in *V. cholerae*. Complementation experiments, with either *V. cholerae* RecA (RecA*_Vch_*) or *E. coli* RecA, showed a restoration of the WT frequency, indicating that this activity is not specific to the RecA*_Vch_*. In the lexAind- strain, in which the SOS response is inhibited (non-inducible LexA derivative [38]), we did not observe any significant change in recombination frequency, meaning that the RecA*_Vch_* effect in integron recombination is independent of the SOS response. We also performed conjugation assays testing the effect of RecA*_Vch_* on the recombination between *attC* sites (i.e., using an *attIA* deleted *V. cholerae* strain). In this case, we did not observe any effect of the RecA*_Vch_* protein, meaning that the effect of RecA*_Vch_* is specific of the *attIA* × *attC* recombination. From all these results, here we discuss, step by step, the potential mechanism of RecA*_Vch_* action.

#### 5.2.3. RecA and *attC* Site Folding

What is well known is that SSB, by wrapping ssDNA in multiple DNA binding modes, can diffuse along DNA to remove secondary structures and ensure the RecA elongation filament [90]. This SSB effect was previously proven on folded *attC* sites in the absence of integrase, allowing one to preserve *attC* site integrity (see above, [77]). In the same way, one hypothesis would be that RecA*_Vch_* would protect single-stranded DNA, such as *attC* sites, which are prone to nuclease degradation [91]. Nevertheless, we did not observe any influence of RecA*_Vch_* on *attC* × *attC* reaction mediated by IntIA*_Vch_*, while this reaction also requires ss *attC* site integrity and folding [56]. Moreover, during our replicative assay, in which *attC* sites are delivered on a double-stranded DNA molecule, we still observed an important effect of the RecA*_Vch_* protein. Indeed, whatever the way we used to deliver *attC* sites, i.e., the single-stranded (conjugation and replicative assays) or the double-stranded (replicative assay) ways, we observed an important effect of the RecA*_Vch_* protein on *attIA* × *attC* recombination [56]. Altogether, these results show that the RecA*_Vch_* effect is not linked to the regulation of the *attC* site folding during the first step of recombination.

#### 5.2.4. RecA and aHJ Replicative Resolution

The *attIA* × *attC* synaptic complex formed during the recombination could impede the aHJ resolution by blocking replication fork progression. The effect of RecA would be then to destabilize this synaptic complex, leading to the final product. RecA could bind ss *attC* sites and help to maintain the *attC* site in a non-folded single-stranded form favoring the replicative resolution step. SSB would also act at this step by flattening the folded *attC* sites, as demonstrated in Loot et al. 2014 and allowing RecA binding. This formed RecA/unfolded ss *attC* site structures could be assimilated to a nucleofilament in the homologous recombination process. Note that we did not observe this RecA*_Vch_* effect on *attC* × *attC* recombination, suggesting a differential configuration between both *attC* × *attC* and *attIA* × *attC* complexes. Indeed, synapses formed during both reactions catalyzed by IntI1 are known to be different since they involve, respectively, two flexible bs *attC* sites, or a flexible bs *attC* and a stiffer ds *attI* [13]. This is also the case for the *attC* × *attC* and *attIA* × *attC* synaptic complexes formed by the IntIA integrase. Such differences explain why synapse architecture for the *attIA* × *attC* reaction versus *attC* × *attC*, which may potentially require the assistance of differential accessory proteins. In line with this hypothesis, our results, like those previously published, show that the recombination reactions involving IntIA*_Vch_* synaptic complexes are impacted differently when the *V. cholerae* integron platform is transferred in *E. coli*. Indeed, when IntIA is expressed in *E. coli*, while the *attC* × *attC* reaction can be catalyzed, productive *attIA* × *attC* reactions are almost undetectable in this host [92]. Thus, indicating that differential host factors seem to be required for both reactions, and some of them are only required for the cassette insertion in *attIA* mediated by IntIA. All together, these observations lead us to propose the model in Figure 7.

#### 5.2.5. RecA and aHJ Reparation

RecA*_Vch_* could also possibly act to repair damages (ds break or ss cleavages) that could be generated by DNA constraints exerted by the *attIA* × *attC* synaptic complex. These damages could occur just downstream of this synaptic complex, blocking the replication fork and impeding the aHJ resolution. The RecA mechanism of action would be similar to its known repair role on stalled replication forks [93,94,95,96,97]. Again, this supposed RecA mechanism of action would be specific to the *attIA* × *attC* reaction versus the *attC* × *attC* reaction catalyzed by IntIA*_Vch_*, meaning that constraint-generating damages would be differential between both synaptic complexes.

Another hypothesis is that the RecA*_Vch_* protein could act on the replicative mechanism itself, which participates in the resolution of the aHJ. Even if not demonstrated, it was proposed that the aHJ formed between two *attC* sites will be resolved by a replicative pathway, as for *attI* × *attC* reaction [36]. However, we do not favor this hypothesis since it is likely that the aHJ resolution mechanism is the same for both reactions and that we will observe a RecA*_Vch_* impact only on the *attIA* × *attC* reaction catalyzed by IntIA*_Vch_*.

## 6. Evolutionary Success of MI: A Matter of Host Factors

### 6.1. Specific Host Factors Are Involved in the Cassette Recruitment in SCI but Not in MI

MI and SCI are regulated by a panel of common host factors. For example, the regulation of the integrase expression by the SOS response involves the same host factors for both SCI and MI since LexA binding boxes are found in both SCI and class 1 MI P_int_ promoters. The integrase expression is therefore dependent on the RecA protein, which forms a nucleofilament triggering the SOS response. While RecA is critical for the cassette recruitment in the *V. cholerae* SCI, RecA is not involved in the *attI**1* × *attC* reaction mediated by IntI1, the integrase of the class 1 MI [56,77]. This suggests that this RecA effect is specific to the *V. cholerae* SCI and that, among host factors, some may be different between SCI and MI. We performed assays in *E. coli* and *V. cholerae* using the IntI1 integrase and demonstrated that whatever the reaction (*attI1* × *attC* or *attC* × *attC*), the integron system works similarly in both bacterial species. This is not the case for the SCI of *V. cholerae*. Indeed, we observed that IntIA recombines 2000 fold less efficiently *attIA* × *attC* reaction, when expressed in a heterologous host such as *E. coli* [92]. The simplest explanation for the observed low level of integration in the in vivo reconstituted system in *E. coli* is that IntIA, unlike IntI1, requires one or more accessory proteins for the integration process, which is either absent or too divergent in *E. coli* to sustain the IntIA-mediated integration process. We failed at recovering *attIA* × *attC* recombination by expressing RecA*_Vch_* in *E. coli*, suggesting that the RecA*_Vch_* protein is not the missing factor in *E. coli* impeding this reaction to efficiently occur. This is not surprising since the RecA*_Ec_* and RecA*_Vch_* present a very high level of identity (80%) and the RecA*_Ec_* is able to complement the RecA*_Vch_* in *V. cholerae* both for its classical recombination function and for the *attIA* x *attC* reaction. Therefore, we hypothesize that one or more specific *V. cholerae* host factors are missing in *E. coli*. These results demonstrate that at least two host factors, RecA and another unknown factor, are involved in the *attIA* × *attC* recombination in *V. cholerae*.

### 6.2. The Evolutionary Success of MI Could Be Linked to a Specific Host Factor Independence

It is tempting to speculate that the success of class 1 MI, carried on broad host-range plasmids and operational in multiple hosts, could be due to an IntI1 activity independent of specific host factors and dependent only on universal clues (Figure 8). This could explain why IntI1 is able to recombine at the same rate in *E. coli* and *V. cholerae* (see above). Classes 2 and 3 MI have also been shown to be functional in *E. coli* [98], and one can consider that the HF independence of class 1 MI holds true for these as well. This differential HF recruitment between MI and SCI is consistent with the observed widespread MI dissemination among bacterial species and their evolutionary success. It has been proposed that *attI* sites have co-evolved with their cognate integrases since cross recombination assays between non-cognate *attI*/*IntI* partners are generally inefficient. We suggest that the success of class 1 MI dissemination could be the result of this co-evolution between the integrase IntI1 and its cognate *attI1* site, allowing cassette recombination to take place in absence of specific host factors. The high level of divergence between IntIA*_Vch_* and IntI1 (only 45% of identity [9,13]) and the lack of identity between *attIA* and *attI1* sites likely reflect the observed differences in their functioning. Indeed, *attI1* contains two supplementary binding sites, the direct repeats (DR1 and DR2), while the *attIA* site does not present such repeated sequences. While these DR are dispensable for the *attI1* × *attI1* reaction, it has been shown that their presence favors the *attI1* × *attC* reactions [99]. DR of *attI1* sites may have been acquired and used as topological filters [99], thus replacing host factors that regulate supercoiling. However, dissemination of MI seems limited to the Gram-negative sphere. Indeed, only a few MI have been found in Gram positive bacteria [100,101]. HF are hypothesized to impact integron ability to disseminate between distantly related species, for example, causing the important distribution bias between Gram negative and Gram-positive bacteria. This is surprising since both types can share the same environment and susceptibility to some antibiotics, an ideal situation for integron expansion. As mobile elements can be exchanged between them [102], MI would supposedly be able to spread in both groups. The yet unknown HF necessary for integron functioning could be involved in this discrepancy. In Gram positive bacteria, these HF would be too divergent from their functional counterparts (or even be simply absent) to allow integron functioning. The integron system would then be dormant, and its presence would be more costly than beneficial. Integron spread would thus be counter-selected, leading to this biased distribution.

## 7. Conclusions

Integrons are an exquisite model for the understanding of the evolutionary compromises MGE have to face. Indeed, they can minimize their burden on their hosts while ensuring that they can provide a measurable benefit when needed (in stress conditions), but this comes at a cost: an intimate connection with the physiology of their regular host. Alternatively, they can evolve more selfishly toward a strategy of expansion by spreading in multiple hosts, though with a suboptimal functioning cost in most of these possible hosts. Integrons seem to have explored these two antagonistic evolutionary behaviors and selected two subfamilies able to provide different selective benefits: the sedentary chromosomal integrons and the mobile integrons. Evidence suggests that mobile integrons evolved from SCI, but this evolution came at a cost: the relative emancipation from the several host factors that couple cassette capture and recombination to the host needs and limit their burden. The only conserved host factor for the functioning of both integron families is the LexA protein, and its essential role for the integrase expression control. Host factors have a dual role in the integron system. First, they ensure a link between bacterial physiology and the integron recombination, coupling the event of cassette shuffling when bacteria need to evolve. However, secondly, they could be a limiting factor for the mobilization of integrons and their dissemination in remotely related bacterial species, if they are too specific and/or missing from these bacteria. From what we reviewed here, one can clearly see the inverse correlation between the degree of host factor dependence and specificity, and the MGE aptitude for trans-genera dissemination. Any interconnection between the integron and the host seems to come at a cost of constraints in terms of dissemination. However, we should emphasize that the two integron families have kept a common recombination process and are able to recombine a large fraction of the *attC* sites, showing that they have retained an interdependence. Indeed, if the MI probably owe their high success to their limited need of specific HF, they however rely on the SCI cassette reservoirs to provide them with an unlimited source of cassettes to spread unbridled. Conversely, MI may bring antibiotic resistance cassettes to SCI, in a time where bacteria are confronted to an adaptive response to an extent and a time frame that they may have never met before the modern era.

## Figures and Tables

**Figure 1 cells-11-00925-f001:**
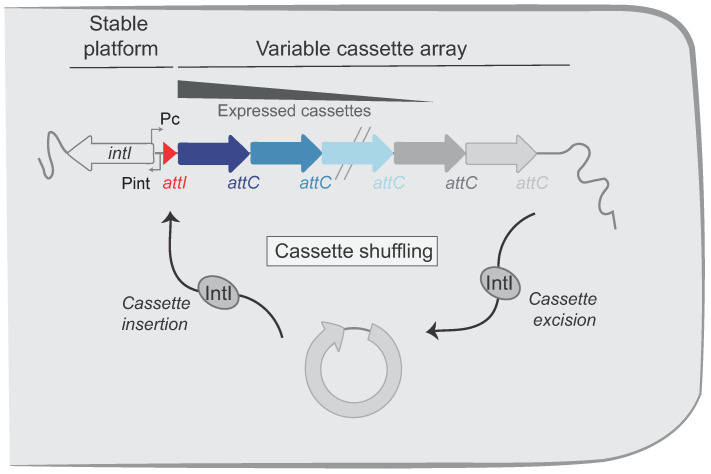
Organization of the integron system. Integrons include a stable platform and a variable cassette array. The platform is composed of the integrase gene (*intI*) under the control of the integrase promoter (P_int_) and encoding the integron integrase (IntI). The platform also contains the cassette promoter (P_C_), which directs transcription of proximal cassettes and the *attI* recombination site in which cassettes are inserted. The cassettes in the variable cassette array are represented by successive arrows. Their color intensities reflect their level of expression. Integrase catalyzes both excision (*attC* × *attC*) and insertion (*attI* × *attC*) of cassettes, leading to cassette shuffling.

**Figure 2 cells-11-00925-f002:**
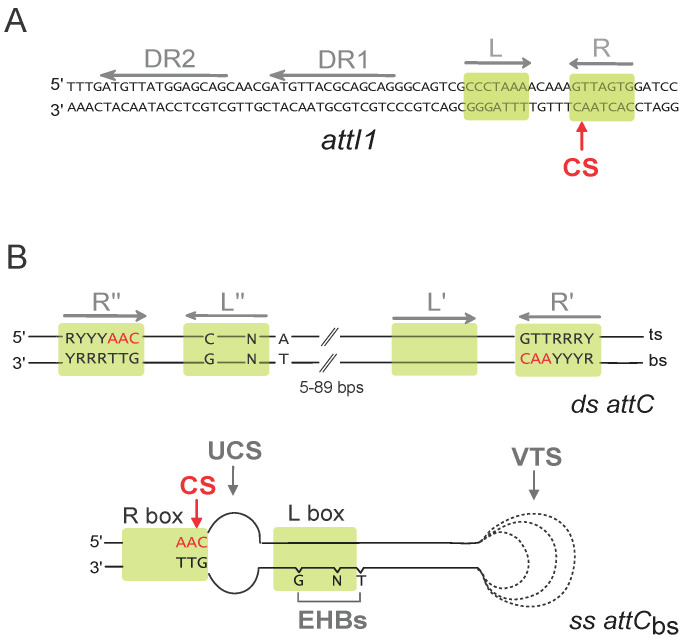
*att* recombination sites. (**A**) Sequence of the double-stranded *attI1* site. Direct repeats (DR1 and DR2), and both left and right (L and R) imperfect inverted repeats are indicated by grey arrows. Putative integrase binding sites are represented by green boxes. The cleavage site (CS) is indicated by a red arrow. (**B**) Schematic representation of the double-stranded (ds) and single-stranded (ss) bottom strand (bs) *attC* sites. Only the conserved nucleotides are indicated. Inverted repeats (R″, L″, L′, and R′) are indicated by grey arrows. Green boxes show putative IntI binding sites, and the cleavage site (CS) is indicated by a red arrow. R, purine; Y, pyrimidine; N, any bases; bs, bottom strand; and ts, top strand. For the *attCbs*, the structural features, namely, the unpaired central spacer (UCS), the extrahelical bases (EHBs), and the variable terminal structure (VTS), are indicated. The dotted lines represent the VTS length variability among *attC* sites.

**Figure 3 cells-11-00925-f003:**
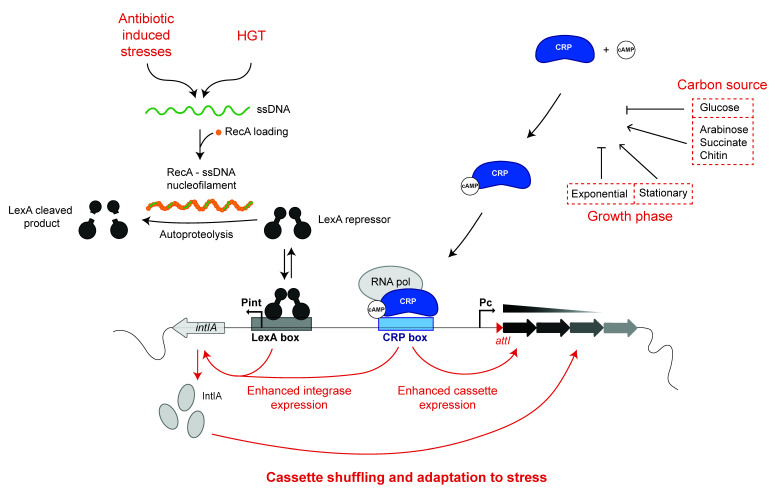
Regulation of the *Vibrio cholerae* SCI promoters. The *intIA* and cassette promoters are regulated by several processes. The main triggering signal for integrase expression is the bacterial SOS response. Indeed, the integrase expression is repressed in normal conditions by the binding of the LexA repressor on the P_int_ LexA box. In stress conditions (genotoxic stress) or in conditions of horizontal transfer of single-stranded DNA (ssDNA), the RecA- ss nucleofilament is formed and induces the autoproteolysis of the LexA repressor, and the integrase is expressed. Both integrase (P_int_) and cassette (P_C_) promoters are regulated by the catabolic repression since both share a CRP box. For instance, in presence of glucose, the P_int_ and P_C_ promoters do not efficiently function.

**Figure 4 cells-11-00925-f004:**
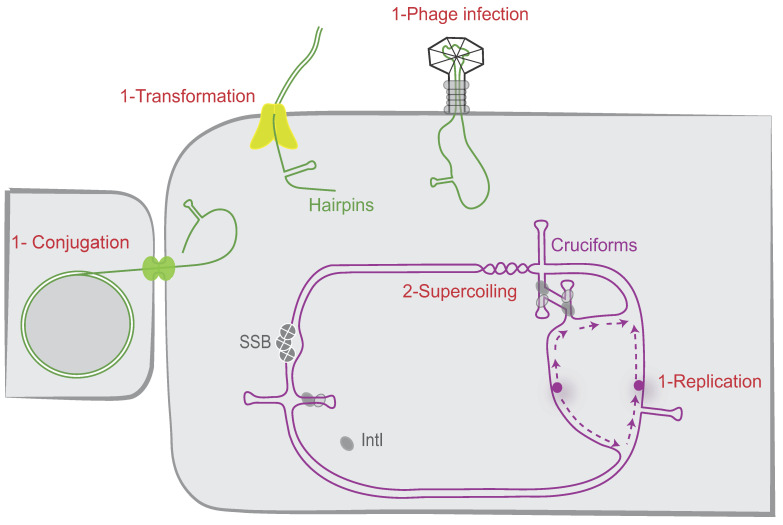
*attC* site folding pathways. The different possible cellular pathways allowing proper folding of the *attC* sites are shown. The *attC* sites can be folded during the single-stranded DNA pathway (1) when delivered during replication, conjugation, phage infection, and transformation. They can also be folded from supercoiled double-stranded DNA (2) as cruciform structures. IntI and SSB monomers are, respectively, represented as grey ovals and grey circles. The origin of replication is represented by a purple circle and the newly synthesized leading and lagging strands by dashed purple lines.

**Figure 5 cells-11-00925-f005:**
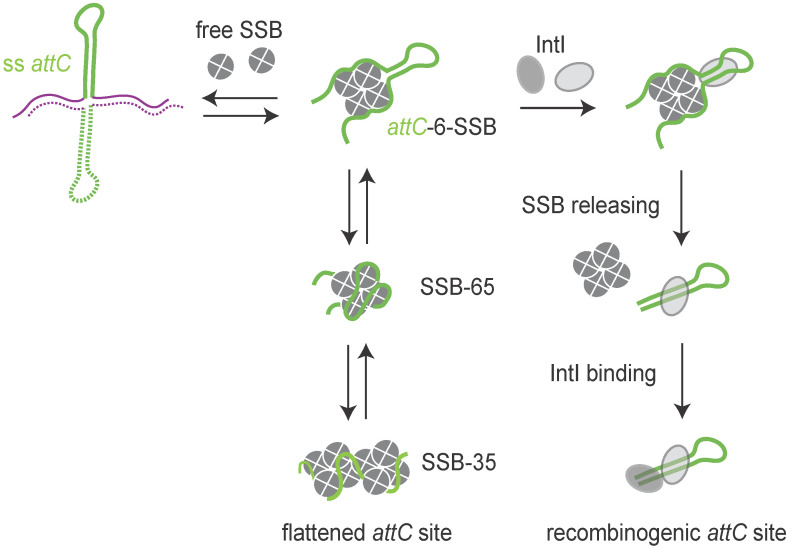
Model of the SSB protein flattening the folded *attC* sites. The model shows how SSB opens the *attC* hairpin via the *attC*-6–SSB state, followed by the SSB-65 and SSB-35 modes, ensuring the integrity of the *attC* sites in absence of integrase (IntI). The integrase acts by capturing the *attC* sites at the moment of their extrusion, preventing the melting effect of SSB. Thus, SSB is released and the *attC* folded structure is stabilized.

**Figure 6 cells-11-00925-f006:**
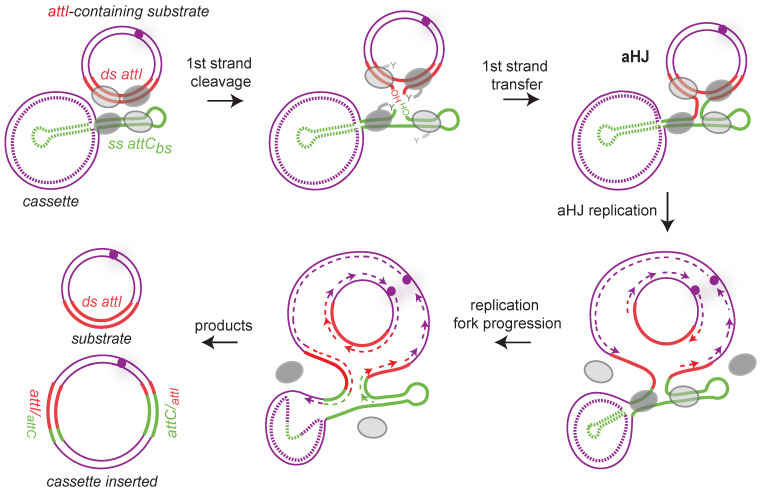
Replicative resolution pathway during cassette insertion in *attI* site. Both substrates (*attI*-containing replicon and the cassette) are presented by purple lines, and the single-stranded *attC* bottom strand (ss *attC_bs_*) and double-stranded *attI* (ds *attI*) sites are represented, respectively, by green and red lines. Note that the top strand of the *attC* site is represented as a dotted line because we do not know the nature of the cassettes (ss or ds). Synaptic complexes during the first strand cleavage and transfer are shown. Only one strand from each duplex is cleaved and transferred forming an atypical Holliday junction (aHJ) due to the single-stranded nature of the *attC* site. Each time, the four integrase protomers are shown. The two activated protomers are represented by dark gray ovals and the inactive ones by light gray ovals. The aHJ resolution implies a replication step. The origin of replication is represented by a purple circle and the newly synthesized leading and lagging strands by dashed purple lines. Both products are represented: the inserted cassette results from the bottom strand replication and the initial *attI*-containing replicon from the top strand replication. Hybrid *attC* and *attI* sites are indicated.

**Figure 7 cells-11-00925-f007:**
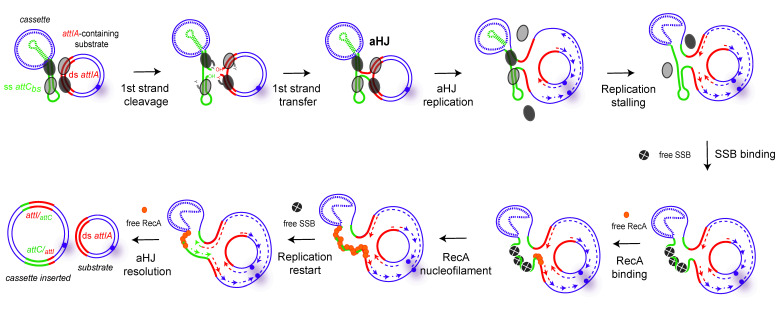
Model of the RecA mechanism of action in *Vibrio cholerae* during cassette insertion in *attIA* site mediated by IntIA. The first steps are the same as in Figure 3, up to the formation of the atypical Holliday junction (aHJ). After that, the model suggests an arrest of the replication fork due to the structure of the aHJ. In order to ensure the replication fork restart, we propose the intervention of both SSB (tetramers of SSB, grey circles) and RecA (orange circles) proteins. SSB-assisted RecA filament growth on hairpin DNA would allow for the replication fork to restart and resolve the aHJ. The SSB protein can bind the *attC* site in one of two binding modes, SSB-35 or SSB-65, but here we represented the SSB-35 mode.

**Figure 8 cells-11-00925-f008:**
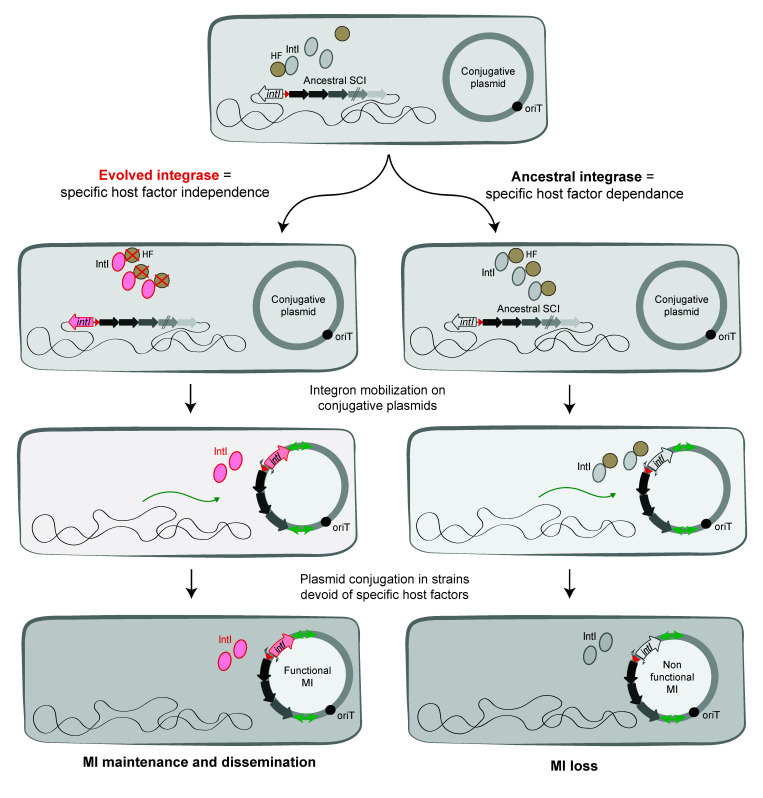
Evolution of integrons toward a limited host factor dependence. The figure shows the evolution of integrases followed by the capture of sedentary chromosomal integrons (SCI) by transposons and their mobilization on conjugative plasmids. The captured SCI are transferred by conjugation in bacteria and now called Mobile Integrons (MI). Their maintenance and dissemination in new bacteria reside in their capacity to have evolved limiting their specific host factor (HF) dependence.

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
