# Peer review of "Unbridled Integrons: A Matter of Host Factors"

_cells, 2022, doi:10.3390/cells11060925_

Round 1

Reviewer 1 Report

The manuscript entitled “Unbridled integrons: a matter of host factors” describes in depth the intertwining of integrons and the physiology of their hosts, through a detailed description of the regulatory networks and proteins that connect them. The manuscript is well structured and written, and is exhaustive in the revision of the literature related to the subject. 

The authors provide a view on integron recombination that fits well the themed issue in honor of Prof. Radman. I believe the review is sound and there are no major issues to highlight. This is not surprising since the authors are the leading group in the field, especially in these aspects of integron recombination. I simply have a couple of comments on some hypotheses discussed in the paper I do not necessarily agree upon. I would like to comment them, in case the authors would want to clarify or build upon in their paper.

Abstract: Line 22: “the progressive loss of host factors”. This comment is linked to the one on lines 773-779 (see below). I am not sure there is evidence of the evolutionary trajectory that integrons have followed. It could be from more to less connected to the host or viceversa. I think it is safer to say that among extant SCIs, less connected SCIs may have been more successful upon mobilization than more connected SCIs. Or in other words: those connected to more universal cues (like the SOS response) are more successful than those tied to species specific cues that are more life-style dependent (like CRP).

Lines 542-544: The reason why CDS in cassettes are shorter than in the rest of the replicon is not clear. Specifically, because after mentioning the limitations in recombination when attC sites are at distances longer than Okazaki fragments, the authors claim that this is not the reason for the shorter CDSs, but that the constraint is linked to cassette genesis. Could you please develop a little further the rationale?

Lines 773-779. On the co-evolution of class 1 integrases and attI1 sites upon mobilization. Given the high degree of conservation of class 1 integrons on mobile plasmids it seems odd that the co-evolution process would have taken place after mobilization by transposons. Had this been the case, one would expect to find many integrons with non-identical sequences, and mutations would necessarily be found at attI sites and/or the integrase. The only major variation in sequences of class 1 integrons are those derived from Pc variants, which proves the case that variation can be found in class 1 integrons when selection takes place, but these mutations are a priori unrelated to what the authors mention. To the best of my knowledge there are no such attI1/intI1 variants (attI1 sites without DRs, for instance, as suggested in the text). Hence, it would seem more likely that any evolution of class 1 integrons towards a state less connected to the host, likely happened before transposition to plasmids. Maybe this is the reason that of all mobilized integrons class 1 is the most successful; but it likely occurred before mobilization.

In previous articles from the authors and others some discrepancies between 16S and intI phylogenies have been shown for Vibrio fischeri, hinting at HGT over long evolutionary periods (reviewed in Mazel, 2006). It could be an interesting case to mention, and to complement the example of mobile integrons. For instance whether or not LexA boxes and, specially, CRP boxes are present in V. fischeri integron, and if this is also the case in its closest relatives (Microbulbifer degradans or Shewanella spp. according to Nemergut et al 2004). i.e.: if integron mobilization over long evolutionary periods has only been detected among platforms bridled similarly, or if not, whether the lacking connections to the new host appear afterwards.   

Other than these comments -that I hope serve to enrich the discussion- I believe this review is an excellent one, that addresses exhaustively the intertwining between host and integrons at a molecular and mechanistical level.

Author Response

The manuscript entitled “Unbridled integrons: a matter of host factors” describes in depth the intertwining of integrons and the physiology of their hosts, through a detailed description of the regulatory networks and proteins that connect them. The manuscript is well structured and written, and is exhaustive in the revision of the literature related to the subject. 

The authors provide a view on integron recombination that fits well the themed issue in honor of Prof. Radman. I believe the review is sound and there are no major issues to highlight. This is not surprising since the authors are the leading group in the field, especially in these aspects of integron recombination. I simply have a couple of comments on some hypotheses discussed in the paper I do not necessarily agree upon. I would like to comment them, in case the authors would want to clarify or build upon in their paper.

Abstract: Line 22: “the progressive loss of host factors”. This comment is linked to the one on lines 773-779 (see below). I am not sure there is evidence of the evolutionary trajectory that integrons have followed. It could be from more to less connected to the host or viceversa. I think it is safer to say that among extant SCIs, less connected SCIs may have been more successful upon mobilization than more connected SCIs. Or in other words: those connected to more universal cues (like the SOS response) are more successful than those tied to species specific cues that are more life-style dependent (like CRP).

We rephrased the abstract to take this into account.

Lines 542-544: The reason why CDS in cassettes are shorter than in the rest of the replicon is not clear. Specifically, because after mentioning the limitations in recombination when attC sites are at distances longer than Okazaki fragments, the authors claim that this is not the reason for the shorter CDSs, but that the constraint is linked to cassette genesis. Could you please develop a little further the rationale?

To simplify and shorten the review, as requested by rev. 2 and 3, we decided to remove this part as it was not directly in line with the context of our review.

Lines 773-779. On the co-evolution of class 1 integrases and attI1 sites upon mobilization. Given the high degree of conservation of class 1 integrons on mobile plasmids it seems odd that the co-evolution process would have taken place after mobilization by transposons. Had this been the case, one would expect to find many integrons with non-identical sequences, and mutations would necessarily be found at attI sites and/or the integrase. The only major variation in sequences of class 1 integrons are those derived from Pc variants, which proves the case that variation can be found in class 1 integrons when selection takes place, but these mutations are a priori unrelated to what the authors mention. To the best of my knowledge there are no such attI1/intI1 variants (attI1 sites without DRs, for instance, as suggested in the text). Hence, it would seem more likely that any evolution of class 1 integrons towards a state less connected to the host, likely happened before transposition to plasmids. Maybe this is the reason that of all mobilized integrons class 1 is the most successful; but it likely occurred before mobilization. Rev 1 is right and we rephrased the text and corrected the figure 8 and its legend to reflect this point.

In previous articles from the authors and others some discrepancies between 16S and intI phylogenies have been shown for Vibrio fischeri, hinting at HGT over long evolutionary periods (reviewed in Mazel, 2006). It could be an interesting case to mention, and to complement the example of mobile integrons. For instance whether or not LexA boxes and, specially, CRP boxes are present in V. fischeri integron, and if this is also the case in its closest relatives (Microbulbifer degradans or Shewanella spp. according to Nemergut et al 2004). i.e.: if integron mobilization over long evolutionary periods has only been detected among platforms bridled similarly, or if not, whether the lacking connections to the new host appear afterwards.  Phylogenetic analysis of attC sites found in integrons have been performed (Rowe-Magnus et al 2001 and 2003; Boucher et al, 2006) and revealed the presence of three distinctive clades (Vibrio cholerae, vulnificus and fischeri) meaning that, if HGT occurred between SCI, it was before the speciation of these strains or between more closely strains inside a same clade. Furthermore, it will be very speculative to discuss CRP in remote species as CRP and its binding sites have not been characterized in these species. We think it is not necessary to include these considerations in this review to avoid complicating and confusing the reader.

Other than these comments -that I hope serve to enrich the discussion- I believe this review is an excellent one, that addresses exhaustively the intertwining between host and integrons at a molecular and mechanistical level.

Reviewer 2 Report

Unbridled integrons: a matter of host factors

 Egill Richard, Baptiste Darracq, Céline Loot and Didier Mazel

Comments for the Authors

Integrons are powerful recombination platform that allow bacteria to evolve and adapt rapidly to environmental conditions through dynamic capture, rearrangement and expression of genes carried in so-called cassettes. They are major players in the spread of multidrug resistance among clinical pathogenic bacteria and therefore represent a serious threat to public health. Two types of integrons are known: mobile integrons found on conjugative plasmids and sedentary chromosomal integrons. Both types share a similar organisation including the integrase gene under the control of its promoter, the integration site and the cassette promoter driving the expression of the genes encoded in the cassette array downstream. The review focuses on the interdependence of two types of integrons and also their connection with the host physiology, in particular the multiple controls by host factors on integrase expression, substrate folding, and resolution of recombination intermediate…. This differential trade-off between both types of integrons and host assures their successful spread and cohabitation. 

The review is reasonably structured and generally well written. It is all the way through documented by crucial data mostly obtained in Mazel labs. However, it is preferable to homogenize writing style throughout the manuscript and improve figures (resolution of drawing and legend). Some sections need sub-heading to facilitate reading.

Other points:

P2

 Figure 1 legend: dot is missing, attI and attC sites should be better shown by colour and drawing respectively

P3

Figure 2: quality is mediocre

L111: the block “R, purine; Y, pyrimidine; N, any bases; ts, top strand” should include also bs and to be moved to L108  

P5

 L151: integrase expression regulation

L180: what does it mean “gradient of expression”?

P6

 Figure 4: The title could be rather “regulation of SCI promoters”? intIA is neither mentioned in the legend nor in the text. If the objective is to illustrate general regulation of integrons promoters, a second drawing of MI would be very useful

L194: , the integrase is expressed

L197-208: this paragraph needs to be reconsidered including clear statement about MI and Pc types.

L203: Without explanation here (I found it later in the text), it’s difficult to understand why Pc can confer different recombination properties to IntI and lead to amino acids substitutions in intI gene

P7

L226: come from double-strand break repair

L232: translesion/al replication system

L246: A sentence is needed to state that while the regulation by SOS response described above is also relevant to MIs but exception exists

P8

L291-292: I would like authors to develop this point

L299: be rearranged at random

L317: It would like to have clarification about what is present on this plasmid.

P9

L341: A reference/review on conjugation is needed here

P10

The paragraph on transformation is too long, it is preferable to divide it into shorter ones and add sub-headings

L395: As mentioned above, IntIA never introduced before

L430 and later on: to cite Figure 4 when describe Pc and Pint in MI and SCI

P11

L436: forms biofilms on the shells of crustaceans…

L479: possible involvement in/with the attC structuration

L431-455: The first paragraph needs reference on induction of transformation in V. cholerae by chitin and other factors

P12

L498: Figure 5 legend: IntI is represented but not SSB

L506-507: ssDNA is unwound from the duplex plasmid, actively transferred into a recipient neighboring bacterium and recircularized

L509: Upon conjugation, the recombination frequencies are the same regardless of VTS lengths. What is about frequencies in other situations?  I think that a short paragraph describing ss structured cassettes stability is necessary to introduce this point

P13

L512: the cassette recruitment is also facilitated by ….

L528-529: I would say: This has been proven by cloning an attC site in both orientations toward replication in a unidirectional replicated plasmid and comparing their respective integration rate

L522-551: Again, this paragraph is too long

L546: a correlation between the cassette arrays orientation and their respective sizes

L544-550: What is known about the origin of this discrepancy between Vibrio and other species? 

L554: I would say here to introduce RecA: proteins able to bind single-stranded DNA as SSB and RecA. If authors are focusing only on SSB here, RecA should be removed

L563: I would say: Thus, the integrase is able to counterbalance…

P15

L614: It’s preferable to extend the phrase to add mechanism leading to change in DNA topology

L628: the host machinery of replication or replication machinery

L646: connecting the integron with the cell physiology.

P16

L652: The nomenclature should be kept throughout the sentence: RecAVch and ReAEc or V. cholerae RecA and E. coli RecA

Figure 7 is rather tiny and needs to be improved. The cassette top strand could be represented in dotted line and lighter colored   

L678-688: There is no reference for these experiments

P17

L697: What is known about in vitro effect of RecA on attIA x attC complex/recombination?

L706: Sentence is truncated

L741: IntIA, recombines 2000 fold less efficiently attIA x attC reaction

P18

L755: I would say “The evolutionary success of MIs could be linked to the limited host factors dependence”

L768: Giving % of identity/conservation between IntIAVch and IntI1 would be appreciable

L776: DRs of attI1 sites may have been acquired

L779: Is there any experience to introduce Integron into Gram+ bacteria?

Again, the paragraph is too long, it’s better to divide it into shorter ones with sub-headings

P20

L816: one can clearly see

L982: Reference is incomplete

Author Response

Unbridled integrons: a matter of host factors

Comments for the Authors

Integrons are powerful recombination platform that allow bacteria to evolve and adapt rapidly to environmental conditions through dynamic capture, rearrangement and expression of genes carried in so-called cassettes. They are major players in the spread of multidrug resistance among clinical pathogenic bacteria and therefore represent a serious threat to public health. Two types of integrons are known: mobile integrons found on conjugative plasmids and sedentary chromosomal integrons. Both types share a similar organisation including the integrase gene under the control of its promoter, the integration site and the cassette promoter driving the expression of the genes encoded in the cassette array downstream. The review focuses on the interdependence of two types of integrons and also their connection with the host physiology, in particular the multiple controls by host factors on integrase expression, substrate folding, and resolution of recombination intermediate…. This differential trade-off between both types of integrons and host assures their successful spread and cohabitation.

The review is reasonably structured and generally well written. It is all the way through documented by crucial data mostly obtained in Mazel labs. However, it is preferable to homogenize writing style throughout the manuscript and improve figures (resolution of drawing and legend). Some sections need sub-heading to facilitate reading.

Other points:

P2

 Figure 1 legend: dot is missing, attI and attC sites should be better shown by colour and drawing respectively

DONE

P3

Figure 2: quality is mediocre

DONE. We replaced all the figures by new high quality .png figures

L111: the block “R, purine; Y, pyrimidine; N, any bases; ts, top strand” should include also bs and to be moved to L108 

DONE

P5

 L151: integrase expression regulation

DONE

L180: what does it mean “gradient of expression”?

We added a new sentence

P6

 Figure 4: The title could be rather “regulation of SCI promoters”? intIA is neither mentioned in the legend nor in the text. If the objective is to illustrate general regulation of integrons promoters, a second drawing of MI would be very useful

We changed the title as suggested by the rev 1 and we also mentioned IntIA.

L194: , the integrase is expressed

DONE

L197-208: this paragraph needs to be reconsidered including clear statement about MI and Pc types.

We slightly modified the text but we did not reconsider this paragraph since PC promoter regulation by HF in MIs is poorly known and documented (apart by LexA). Moreover, a specific review describing all the PC promoters and their variants in MIs has been recently published from Fonseca et al, 2022. So, we added this new reference.

L203: Without explanation here (I found it later in the text), it’s difficult to understand why Pc can confer different recombination properties to IntI and lead to amino acids substitutions in intI gene

DONE, we added an explanation

P7

L226: come from double-strand break repair

DONE

L232: translesion/al replication system

DONE

L246: A sentence is needed to state that while the regulation by SOS response described above is also relevant to MIs but exception exists

We have rewritten this paragraph

P8

L291-292: I would like authors to develop this point

We think that describing this point would indeed be very interesting, but it is outside the topic of our review (Host factors involved in integron system regulation) and would confuse the readers.

L299: be rearranged at random

DONE

L317: It would like to have clarification about what is present on this plasmid.

We simplified and partially removed this part as suggested by the rev 3.

P9

L341: A reference/review on conjugation is needed here

DONE

P10

The paragraph on transformation is too long, it is preferable to divide it into shorter ones and add sub-headings

We simplified this part as suggested also by the rev3

L395: As mentioned above, IntIA never introduced before

We introduced IntIA in the 4.1.1 paragraph

L430 and later on: to cite Figure 4 when describe Pc and Pint in MI and SCI

DONE

P11

L436: forms biofilms on the shells of crustaceans…

This paragraph is devoted to what is known on stringent response and IntI1 expression in E coli. This connection has not been checked for V. cholerae and its SCI integrase, so we did not mention biofilms on the shells of crustaceans, on purpose

L479: possible involvement in/with the attC structuration

DONE

L431-455: The first paragraph needs reference on induction of transformation in V. cholerae by chitin and other factors

As mentioned above, this paragraph is on Class1 MI not Vibrio SCI.  All the references concerning transformation induction by chitin were cited in the § on the SOS and HGT (line 368 to 382, Johnston et al, 2014 and Meibom et al, 2005).

P12

L498: Figure 5 legend: IntI is represented but not SSB

DONE

L506-507: ssDNA is unwound from the duplex plasmid, actively transferred into a recipient neighboring bacterium and recircularized

DONE

L509: Upon conjugation, the recombination frequencies are the same regardless of VTS lengths. What is about frequencies in other situations?  I think that a short paragraph describing ss structured cassettes stability is necessary to introduce this point

We added two sentences to introduce this point.

P13

L512: the cassette recruitment is also facilitated by ….

DONE

L528-529: I would say: This has been proven by cloning an attC site in both orientations toward replication in a unidirectional replicated plasmid and comparing their respective integration rate

DONE

L522-551: Again, this paragraph is too long

Agreed, we removed a paragraph of more than 10 lines.

L546: a correlation between the cassette arrays orientation and their respective sizes

Corrected

L544-550: What is known about the origin of this discrepancy between Vibrio and other species?

This is a very interesting question. Nothing is known about this discrepancy and this precise point is under investigation in the lab. We indeed try to understand what are the constraints that govern the SCI orientation in Vibrio strains. We obtained some interesting results which will be submitted for publication soon.

L554: I would say here to introduce RecA: proteins able to bind single-stranded DNA as SSB and RecA. If authors are focusing only on SSB here, RecA should be removed

We rephrased this part.

L563: I would say: Thus, the integrase is able to counterbalance…

DONE

P15

L614: It’s preferable to extend the phrase to add mechanism leading to change in DNA topology

We disagree, this ios just a conclusion, and we largely described (see line 599 to 614) the factors and the mechanisms leading to change in DNA topology.

L628: the host machinery of replication or replication machinery

DONE

L646: connecting the integron with the cell physiology.

DONE

P16

L652: The nomenclature should be kept throughout the sentence: RecAVch and ReAEc or V. cholerae RecA and E. coli RecA

DONE

Figure 7 is rather tiny and needs to be improved. The cassette top strand could be represented in dotted line and lighter colored 

We agree and are sorry for the very poor quality of the figures provided. We have replaced all figures with high quality ones.

L678-688: There is no reference for these experiments

DONE

P17

L697: What is known about in vitro effect of RecA on attIA x attC complex/recombination?

NOTHING. We have just purified the RecA protein and, in collaboration with Vincent Parissi we will carry out in vitro experiments to try to determine the precise mechanism of action of RecA.

L706: Sentence is truncated

SORRY. We corrected it.

L741: IntIA, recombines 2000 fold less efficiently attIA x attC reaction

DONE

P18

L755: I would say “The evolutionary success of MIs could be linked to the limited host factors dependence”

DONE

L768: Giving % of identity/conservation between IntIAVch and IntI1 would be appreciable

DONE

L776: DRs of attI1 sites may have been acquired

DONE

L779: Is there any experience to introduce Integron into Gram+ bacteria?

Interesting question! We are currently performing experiments in collaboration with the MC Ploy team (Thomas Jové and Olivier Barraud) in order to understand what are the barriers in gram-positive bacteria limiting the integron dissemination. Note that we added two refs concerning the integrons in gram positive bacteria.

Again, the paragraph is too long, it’s better to divide it into shorter ones with sub-headings

We removed 5 sentences in this paragraph

P20

L816: one can clearly see

DONE

L982: Reference is incomplete

DONE

Reviewer 3 Report

Very well written updated review. Minor changes need to be included. My suggestions- (i) Introduction of the review is too long. I will suggest to include the description part of the integron in a separate heading. (ii) Diversity of integron in different species of Vibrios should be included. (iii) SOS response and HGT part is not very relevant to the current title. This section should exclude or reduce.

Author Response

Very well written updated review. Minor changes need to be included.

My suggestions-

  • Introduction of the review is too long. I will suggest to include the description part of the integron in a separate heading. We have re-organized the review. We added a heading. We also shortened the introduction. Indeed, we have moved a paragraph and a figure from the introduction to the part entitled “The replication process is involved in the aHJ resolution”.
  • Diversity of integron in different species of Vibrios should be included.

This part on the diversity of integrons in vibrios is very interesting but very broad and no combination experiments have been carried out outside the V. cholerae strains. Nothing is known about the host factors involved in other vibrios. It would indeed be interesting to clone intIA in other vibrios close and further away from vibrio cholerae and to see the recombination capacity of the vibrio cholerae intIA integrase.

  • SOS response and HGT part is not very relevant to the current title. This section should exclude or reduce.

DONE, we simplified these parts by removing paragraphs